# Factors that influence the administration of tranexamic acid (TXA) to trauma patients in prehospital settings: a systematic review

Helen Nicholson ![ORCID],[1] Natalie Scotney,[2] Simon Briscoe,[3] Kim Kirby,[2,4] Adam Bedson,[2] Laura Goodwin ![ORCID],[1] Maria Robinson ![ORCID],[2] Hazel Taylor,[5] Jo Thompson Coon ![ORCID],[6] Sarah Voss ![ORCID],[1] Jonathan Richard Benger[1]

For numbered affiliations see end of article.

**Correspondence to**
Dr Helen Nicholson;
helen5.nicholson@uwe.ac.uk

## ABSTRACT

**Objective** In the UK there are around 5400 deaths annually from injury. Tranexamic acid (TXA) prevents bleeding and has been shown to reduce trauma mortality. However, only 5% of UK major trauma patients who are at risk of haemorrhage receive prehospital TXA. This review aims to examine the evidence regarding factors influencing the prehospital administration of TXA to trauma patients.

**Design** Systematic literature review.

**Data sources** AMED, CENTRAL, CINAHL, Cochrane Database of Systematic Reviews, Conference Proceedings Citation Index—Science, Embase and MEDLINE were searched from January 2010 to 2020; searches were updated in June 2022. Clinicaltrials.gov and OpenGrey were also searched and forward and backwards citation chasing performed.

**Eligibility criteria** All primary research reporting factors influencing TXA administration to trauma patients in the prehospital setting was included.

**Data extraction and synthesis** Two independent reviewers performed the selection process, quality assessment and data extraction. Data were tabulated, grouped by setting and influencing factor and synthesised narratively.

**Results** Twenty papers (278 249 participants in total) were included in the final synthesis; 13 papers from civilian and 7 from military settings. Thirteen studies were rated as 'moderate' using the Effective Public Health Practice Project Quality Assessment Tool. Several common factors were identified: knowledge and skills; consequences and social influences; injury type (severity, injury site and mechanism); protocols; resources; priorities; patient age; patient sex.

**Conclusions** This review highlights an absence of high-quality research. Preliminary evidence suggests a host of system and individual-level factors that may be important in determining whether TXA is administered to trauma patients in the prehospital setting.

**Funding and registration** This review was supported by Research Capability Funding from the South Western Ambulance Service NHS Foundation Trust and the National Institute for Health Research Applied Research Collaboration South West Peninsula.

**PROSPERO registration number** CRD42020162943.

## STRENGTHS AND LIMITATIONS OF THIS STUDY

⇒ Strengths of this review include the extensive searches, the use of best practice methods and tools, reporting according to Preferred Reporting Items for Systematic Reviews and Meta-Analyses guidelines and the input of a multi-disciplinary study team.

⇒ Generalisability was enhanced by the separation of civilian and military studies.

⇒ The review was limited by the small number of relevant studies, which precluded a formal synthesis as well as an English language restriction (two studies excluded due to language).

⇒ In a deviation from the protocol, we were unable to search the International Clinical Trials Registry Platform search portal. However, we were able to search the ClinicalTrials.gov registry which did not identify any new studies for inclusion; we therefore believe the impact of this was minimal.

## INTRODUCTION

At least 20 000 individuals are severely injured in England each year, resulting in approximately 5400 deaths, of which 2400 occur before hospital arrival.[1] The CRASH2 trial provided strong evidence of the effectiveness of tranexamic acid (TXA) in reducing mortality in bleeding trauma patients,[2] with CRASH3 then extending its use to isolated head injuries.[3] When given within 3 hours of injury, TXA reduces death from bleeding in all types of trauma.[4] An additional exploratory analysis concluded that the intervention is more effective if administered as soon as possible, particularly within 1 hour.[5] In their meta-analysis of the CRASH2 and WOMAN trials, Gayet-Ageron *et al* found that immediate treatment improved survival by more than 70%, and that the survival benefit decreased by 10% for every 15 min of delay.[6]

In 2011, South Western Ambulance Service NHS Foundation Trust was the first Emergency Medical Services (EMS) provider (known as ambulance services in England) to implement the recommendations of CRASH2[7]; TXA was subsequently introduced across all UK ambulance services. However, analysis of UK Trauma Audit and Research Network (TARN) data found that only 5% of UK major trauma patients who are at risk of haemorrhage receive prehospital TXA.[8] The percentage of patients given TXA within 1 hour varies, with a reported range of 30%–59%.[9 10]

The timing of TXA administration is the single factor that determines effectiveness,[4 5] and the key determinant of time to administration is the setting in which it is given.[10] TARN data show that when administered prehospital, median time to TXA administration is 49 min, compared with a median time of 111 min when given in hospital.[10]

Determining the full extent of a patient's injuries, and the associated bleeding risk, in the early stages of prehospital care can be challenging. This systematic review aims to examine evidence regarding factors that influence the timely administration of TXA to prehospital trauma patients. Due to the paucity of literature on this subject from UK civilian prehospital practice this review included both international and military-based prehospital care in order to maximise the identification of potential influencing factors. This approach was intended to give the greatest possible insights into TXA administration.

## METHODS
### Searches
We searched: AMED (via EBSCO); CENTRAL (via the Cochrane Library); CINAHL (via EBSCO); Cochrane Database of Systematic Reviews (via the Cochrane Library); Conference Proceedings Citation Index—Science (via Web of Science); Embase (via Ovid) and MEDLINE (via Ovid). The search strategy was developed using MEDLINE by an information specialist (SB) in consultation with the team. It included a sensitivity maximising search filter for identifying studies on EMS staff (paramedics)[11] and the final version was tested using preidentified studies to ensure it retrieved all known relevant research. Searches were conducted on 10 January 2020 with a date limit of 2010 (reflecting the publication date of CRASH-2[2]) and most recently updated on 8 June 2022. Search results were exported to Endnote X8 and deduplicated. The search strategies for each bibliographic database are reported in online supplemental appendix 1.

We supplemented these searches using Clinical-Trials.gov (see online supplemental appendix 1 for search terms), checking reference lists, forward citation searching and OpenGrey (www.opengrey.eu). Forward citation searching used the Science Citation Index (via Web of Science) on 7 July 2020, 29 January 2021 and 13 December 2022. If an included study was not indexed in the Science Citation Index we used either Scopus or Google Scholar. Reference lists of included studies were checked manually.

### Article selection
All primary prehospital research involving trauma patients and detailing any factors influencing TXA administration was included (see online supplemental appendix 2 for inclusion and exclusion criteria). Due to resource constraints the review was limited to English language publications. Study titles and abstracts were screened by two independent reviewers (HN and NS/KK). Disagreements were resolved through arbitration by a third reviewer (SV). The full text of all articles that appeared to meet eligibility criteria were retrieved. Full texts were assessed following the same process as the title and abstract screening. All eligible papers proceeded to data extraction.

### Quality appraisal
The Quality Assessment Tool for Quantitative Studies from the Effective Public Health Practice Project (EPHPP) was independently used by two reviewers (HN and NS/KK) to assess quantitative studies for risk of bias, study design, confounders and results.[12 13] Studies were graded 1–3 (strong, moderate or weak). Qualitative studies were assessed using the Critical Appraisal Skills Programme (CASP) Qualitative Research Checklist.[14]

Studies were not excluded on the basis of poor quality, but methodological limitations and their effect on confidence in the findings of each paper were considered when drawing conclusions.[15]

### Data collection and synthesis
Data were extracted by two independent reviewers (HN and NS/KK) using a predetermined data collection table (see online supplemental appendix 3 for data collection variables). Studies were tabulated and grouped by influencing factors and described with the aim of identifying patterns.

### Patient and public involvement
No patient involved.

## RESULTS
The Preferred Reporting Items for Systematic Reviews and Meta-Analyses (PRISMA) diagram is shown in online supplemental appendix 4. Twenty papers were included in the final synthesis. In order to aid interpretation, the papers were divided into civilian (13 papers) and military settings (7 papers) (table 1, online supplement appendix 5 and table 2, online supplemental appendix 6, respectively). All but one of the papers were quantitative in design. Thirteen of the studies were retrospective observational or service evaluation studies, with four observational cohort studies,[16–19] one case–control study,[20] one case series[21] and one qualitative interview study.[22]

Thirteen studies were rated as 'moderate' using the EPHPP Quality Assessment Tool. One study was rated

**Table 1** Summary table of characteristics of included civilian studies (n=13) (see online supplemental appendix 5 for full table)

| Author, country | Factors influencing TXA administration |
|---|---|
| Bossers et al,[17] Netherlands | Patient age: patients receiving TXA older than those not receiving TXA (47 vs 45 years) |
| | Injury type, severity: patients receiving TXA had higher Injury Severity Score (ISS) (27 vs 26), lower prehospital Glasgow Coma Scale (GCS) score (4 vs 5) and higher heart rate (98 vs 92) |
| El-Menyar et al,[20] Qatar | Resources: patients did not receive TXA if critical care paramedics unavailable |
| Goodwin (2022),[22] UK | Knowledge and skills: inadequate training and a lack of knowledge of the effects of TXA or the evidence base behind its use and a lack of exposure to trauma patients were barriers to its administration |
| | Resources: a lack of time and staffing was a barrier to administration. Helicopter Emergency Medicine Services (HEMS) attendance was a barrier to some as they preferred to wait for a HEMS team member to administer it |
| | Protocol: guidelines felt restrictive or confusing. Disparity between paramedic, HEMS and doctor TXA protocols causes confusion. The drug preparation and administration route were seen as barriers to its use |
| | Consequences and social influences: the benefits of TXA were seen to outweigh the risks. TXA was seen to signal a major trauma patient. Fear of repercussion for administering TXA inappropriately. Opinion of colleagues seen to influence TXA administration |
| | Priorities: TXA not seen as a priority, with administering fluids or distracting injuries often taking precedent. Three-hour window of administration may reduce the perceived urgency of administration. The stress associated with trauma jobs may lead to TXA being overlooked or deprioritised |
| | Injury type, severity: risk of bleeding influenced administration including clinical observations and patient presentation. Participants found identifying patients at risk of bleeding difficult |
| | Injury type, mechanism of injury (MOI): uncertainty over which injuries/conditions TXA is indicated for. MOI and type of injury were part of identify patients at risk of bleeding |
| | Injury type, site: less obvious bleeding including occult/internal haemorrhage was harder to identify |
| Marsden et al,[9] UK | Injury type, MOI: road traffic collisions more likely to receive TXA, falls less likely |
| | Injury type, severity: patients given TXA more likely to have heart rate and blood pressure suggestive of bleeding |
| | Resources: patients given TXA were more likely to be treated by a physician-led crew |
| McQueen and Wyse,[23] UK | Protocol: clinician judgement used to guide administration outside of the protocol |
| | Injury type, MOI: most TXA patients had multiple injuries |
| | Resources: time constraints on scene and the absence of a doctor as part of the HEMS crew meant TXA not given |
| Neeki et al,[16] USA | Resources: not all Emergency Medicine Service providers carrying TXA |
| Ng (2019),[30] Canada | Injury type, severity: more patients receiving TXA had moderate (9–12) or severe (3–8) GCS scores than those not receiving TXA |
| Nutbeam et al,[27] UK | Sex: female patients less likely to receive TXA (OR 0.35, 95% CI 0.33 to 0.36) |
| | Injury type, severity: females less likely than males to receive TXA in all ISS categories |
| | Injury type, MOI: females less likely than males to receive TXA for all injury mechanisms except motor vehicle crashes |
| van Wessem et al,[18] Netherlands | Patient age: patients receiving TXA younger than those not receiving TXA (42 vs 53 years) |
| | Injury type, severity: patients needing prehospital intubation, urgent laparotomy or with more deranged physiology more likely to have received TXA |
| van Wessem and Leenen,[19] Netherlands | Patient age: patients receiving TXA younger than those not receiving TXA (41 vs 51 years) |
| | Injury type, severity: patients receiving TXA slightly more severely injured, had higher Abbreviated Injury Scale (AIS) Head scores and more often prehospitally intubated |
| Vu et al,[21] USA | Priorities: balancing critical interventions, resuscitation and short flight times meant some eligible patients did not receive TXA |
| Wafaisade et al,[32] Germany | Injury type, severity: patients needing prehospital intubation or chest tube placement more likely to receive TXA |
| | Injury type, site: patients with AIS ≥3 for abdomen or extremities more likely to receive TXA. Patients with AIS ≥3 for head or thorax less likely to receive TXA |
| | Age: patients over 60 years old less likely to receive TXA |

Continued

**Table 1** Continued

| Author, country | Factors influencing TXA administration |
|---|---|
| Wong et al,[24] Canada | Age: TXA group notably younger (38.2 years vs 49.1 years) |
| | Injury type, severity: TXA group had higher mean ISS and more patients with hypotension |
| | Resources: more patients receiving TXA had a paramedic of higher certification level in attendance than those who did not receive TXA |

TXA, tranexamic acid.

'strong'[20] due to its case–control design. One abstract[23] was rated 'weak' due to a lack of published information and four other papers were rated 'weak' as a result of study design and data collection methods.[17–19 24] The qualitative study was rated as 'valuable' using the CASP Qualitative Research Checklist.[14] All but five of the studies were retrospective, and data collection was often incomplete. However, all of the studies explained patient dropouts or exclusions except the abstract.[23]

Only two of the papers included paediatric patients in their studies[9 25] and three of the papers limited their studies to adult trauma patients with severe traumatic brain injury.[17–19] Only the Goodwin et al paper had primary outcomes relating directly to factors influencing the prehospital administration of TXA.[22] This was a qualitative study interviewing 18 UK paramedics from one ambulance service about the barriers and facilitators of TXA administration. Using the factors reported from this paper and the reported outcomes from the other papers it was possible to identify several factors of interest: knowledge and skills; consequences and social influences; injury type (including severity, injury site and mechanism of injury); protocols; resources; priorities; patient age; patient sex.

### Injury type

This factor was the most prevalent (14/19 papers)[9 17–19 22–24 26–32] and could be further divided into injury severity, injury site and mechanism of injury.

### Injury severity

Patients with a higher Injury Severity Score (ISS) were more likely to receive TXA in both military and civilian settings.[17 24 29] This was also the case within various subcohorts of combat casualties, such as those who had a tourniquet applied or with gunshot wounds.[29] Civilian patients needing prehospital tracheal intubation[18 19] or chest tube placement were also more likely to receive TXA.[32] However, one paper found no difference in ISS between patients that received prehospital and in-hospital TXA.[9] Betelman Mahalo et al found that TXA administration in combat casualties was associated with analgesic treatment.[26]

**Table 2** Summary table of characteristics of included military setting studies (n=7) (see online supplemental appendix 6 for full table)

| Author, country | Factors influencing TXA administration |
|---|---|
| Benov et al,[25] Israel-Syrian Border | Protocol: TXA for penetrating torso injury regardless of haemodynamic status. All patients receiving freeze-dried plasma with a known time of injury <3 hours received TXA |
| | Injury type, MOI: majority of TXA patients had penetrating injury |
| Fisher et al,[29] Iraq and Afghanistan | Injury type, severity: patients with higher ISS, tourniquet application or serious injuries to the thorax, abdomen, extremities and skin more likely to receive TXA |
| | Injury type, MOI: explosive injury and gunshot wounds were more likely to receive TXA. More likely to have explosive injuries and less likely gunshot wounds in TXA patients who had a tourniquet applied |
| | Patient age: amputation patients receiving TXA were younger than those not receiving TXA (22 vs 25 years) |
| Lipsky et al,[34] Israel | Protocol: 30% of TXA administrations had no clear indication for TXA administration. Altered level of consciousness mistakenly categorised as a sign of haemodynamic instability |
| | Priorities: some non-administrations due to tactical limitations, resuscitation or to avoid delaying evacuation |
| Betelman Mahalo et al,[26] Israel | Injury type, severity: TXA administration associated with analgesic treatment |
| Nadler et al,[33] Israel | Protocol: more conservative protocol in the civilian service compared with the military service but higher proportion of patients given TXA outside of protocol (with clearance) by civilian service |
| Nadler et al,[35] Israel | Protocol: New Clinical Practice Guidelines (indicating TXA at a heart rate of 130 instead of 110) introduced caused a significant decrease in the proportion of TXA administered. Only 22% of patients indicated for TXA received it |
| Tsur (2021)[31], Israel | Injury type, site: isolated neck injuries more likely to receive TXA than no-neck injuries |

ISS, Injury Severity Score; MOI, mechanism of injury; TXA, tranexamic acid.

## Injury site

Injury location influenced TXA administration, with paramedics reporting that less obvious bleeding, such as internal haemorrhage, was harder to identify therefore making the administration of TXA less likely.[22] Combat casualties were more likely to receive TXA if they suffered an isolated neck injury (15%, compared with 4% without neck injuries: p=0.01).[31] This may be due, in part, to the fact that the neck is mostly a non-compressible area, and therefore requires alternative haemorrhage control strategies. However, the number of isolated neck injury patients was small (41, compared with 3185 without neck injuries over a 12-year period).

In one civilian study, when using the Abbreviated Injury Score (AIS) to assess trauma patients, a score of 3 or more for the head or thorax was associated with a significantly lower chance of receiving TXA, whereas for the abdomen or extremities it was associated with an increased chance.[32] However, van Wessem and Leenen found that patients receiving TXA had higher AIS Head scores.[19]

Ng *et al* found that more patients receiving TXA in a civilian setting had a moderately or severely reduced Glasgow Coma Scale (GCS) score compared with those not receiving TXA, although statistical significance was not reported.[30] This was supported by Bossers *et al* who found that patients receiving TXA had a significantly lower prehospital GCS than those not receiving it.[17]

## Mechanism of injury

Patients with multiple injuries made up 72% of patients receiving TXA in one study.[23] Mechanism of injury influenced the likelihood of receiving prehospital TXA in some of the civilian studies. Marsden *et al* compared those receiving prehospital TXA with those receiving it in hospital.[9] Mechanism of injury was a statistically significant factor, with prehospital clinicians more likely to recognise the need for TXA in road traffic collisions than in falls. Penetrating injury was a positive determinant of TXA administration even when protocols did not specifically prompt this.[9 20] Penetrating trauma was also linked to TXA administration in combat casualties,[29 33] with patients sustaining explosive injury and gunshot wounds more likely to receive TXA.[29] Participants in the Goodwin *et al* study confirmed that mechanism of injury played a part in identifying patients at risk of bleeding, however, they also reported uncertainty over which injuries or conditions TXA was indicated for.[22]

## Protocols

This factor encompassed both the contents of protocols (indications for use, route of administration) and how they were used (compliance, confusion) and it was closely linked to injury type. Where described, the details of TXA protocols varied, but usually included exact indications (patient observation ranges and/or mechanism of injury).[18–21 24 25 33–35] Recognised signs of clinical shock were most often the basis of these protocols.[18–21 34] In some, the presence of major trauma,[21] the use of freeze-dried plasma,[25] capillary refill time or altered level of consciousness were also included.[34] However, for the National Israeli Civilian Services only age (>18 years), expected evacuation times (>15 min) and the presence of a non-compressible haemorrhage were specified as indications.[33] The Tactical Combat Casualty Care guidelines used by the US military employ the more subjective indication of 'anticipated to need significant blood transfusion'.[29] This was similar to the 'clinical suspicion of major haemorrhage' or 'clinical judgement' reported in the indications for TXA in some civilian studies.[18 19 23]

This variation in protocol was also mentioned by participants in the Goodwin *et al* study where they felt that the disparity between paramedic, helicopter emergency medical services (HEMS) and doctor TXA protocols caused confusion.[22] They reported that the guidelines for TXA use felt restrictive or confusing, and that the drug preparation and administration route were seen as barriers to its use.

The TXA protocol for the Israel Defence Force specifies use in patients receiving freeze-dried plasma as well as those with penetrating torso injury, regardless of haemodynamic status.[25] It is therefore unsurprising that 50.9% (n=113) who received TXA in the Benov *et al* study presented with penetrating injury and tachycardia, and all patients known to be less than 3 hours from injury who received freeze-dried plasma also received TXA.[25] However, this study was limited by incomplete data collection and the assumption that severely injured patients would likely have died before reaching medical assistance.

## Resources

Time, clinician and TXA availability appear to influence TXA administration.[9 16 20 22 23] In addition to twelve cases (48%) of TXA administrations being outside protocol, McQueen and Wyse noted a further 13 (46%) indicated cases where TXA was not administered.[23] They identified time constraints on scene and the absence of a doctor as reasons for non-administration. El-Menyar *et al* found that the time from EMS activation to hospital arrival was significantly longer in the TXA group (62 vs 74 min, p=0.03).[20] This suggests that TXA administration is associated with increased prehospital time; either due to the requirements of drug administration, or additional procedures that patients who are given TXA require at scene.

The grade of clinician able to administer TXA varied. El-Menyar *et al* reported that TXA could only be administered by critical care paramedics, and suggested that a lack of availability of these staff led to some eligible patients not receiving TXA.[20] Marsden *et al* found that patients who were given TXA were more likely to be treated by a physician-led crew (60.5% vs 31.6%, p<0.001)[9] and Wong *et al* found that patients receiving TXA were attended by a paramedic of higher certification level than those who did not receive TXA.[24] However, some participants in the Goodwin *et al* study reported that HEMS attendance was a barrier to TXA administration as they preferred to wait for a HEMS team member to administer it.[22]

However, the most fundamental resource issue is the availability of the drug itself, and despite strong evidence of effectiveness some services did not carry TXA.[16]

## Priorities

The Goodwin *et al* study found that TXA was not seen as a priority by the paramedics interviewed with fluid administration or the treatment of injuries often taking precedent.[22] Some studies also suggested that other considerations may take priority over TXA administration. For example, Vu *et al* found several patients who met the criteria but did not receive TXA, although the exact number was not disclosed.[21] In these cases, clinicians reported other priorities that required their attention as they balanced critical interventions, resuscitation and short flight times.[21] A similar situation was described by Lipsky *et al* who found that some non-administrations may have been due to tactical limitations, resuscitation or to avoid delaying evacuation.[34] However, they also suggested that raising awareness among staff may prove beneficial.

The participants in the Goodwin *et al* study also suggested that the 3-hour window may reduce the perceived urgency of TXA administration and that the stress associated with trauma patients may lead to TXA being overlooked or deprioritised.[22]

## Patient age

There was some evidence that a patient's age influenced administration, with patients over 60 years less likely to receive TXA in one study[32] and patients receiving TXA being notably younger (38.2 years vs 49.1 years) than those who did not in another study.[24] The van Wessem studies also found that trauma patients with severe traumatic brain injury receiving TXA were younger than those who did not (42 vs 53 years and 41 vs 51 years), although the 7 years of data from the 2021 study were reused but extended by 6 months in the 2022 study meaning these should not be considered separate results.[18 19] However, Bossers *et al* found that patients with severe traumatic brain injury receiving TXA were slightly older than those not receiving it (47 vs 45 years).[17]

In military settings Fisher *et al* found that, of those with amputations, patients receiving TXA were younger than those not receiving TXA (22 vs 25 years, p<0.001),[29] though some of these differences are of questionable clinical significance.

## Patient sex

Nutbeam *et al* was the only paper to report a patient sex difference in TXA administration.[27] They found that female patients in all risk categories and age groups were less likely to receive TXA (OR 0.35, 95% CI 0.33 to 0.36). This difference in receipt of TXA increased with increasing age and decreased as injury severity increased. Women were found to be less likely to receive TXA for all mechanisms of injury except motor vehicle crashes.

## Knowledge and skills

The participants in the Goodwin *et al* study reported that inadequate training and a lack of knowledge of the effects of TXA or the evidence base behind its use were barriers to its administration.[22] A lack of exposure to trauma patients was also reported as a barrier to its administration.

## Consequences and social influences

Paramedics interviewed in the Goodwin *et al* study felt that the benefits of TXA outweigh the risks.[22] However, administering TXA was seen to signal a major trauma patient, leading to concerns about the acceptance of the patient at a major trauma centre or judgement from colleagues if it was given to less severely injured patients. The opinion of colleagues was seen as an influencing factor, with 7/18 paramedics reporting that they check with a senior colleague or hospital staff before administering TXA. Paramedics reported a fear of repercussion for administering TXA inappropriately. They also felt that a lack of visible effects on patients inhibited TXA use.

## DISCUSSION

This review found a paucity of high-quality research addressing the factors that influence prehospital TXA administration. However, despite only one of the included papers having primary outcomes relating directly to factors influencing the prehospital administration of TXA, it was possible to identify several factors from the studies that suggested a host of system and individual-level factors that may be important in determining whether TXA is administered to trauma patients in the prehospital setting. Injury type and protocols were the most prominent factors.

Despite the fact that this systematic review is the most complete to date of factors influencing prehospital TXA administration, there are some limitations. Only two of the studies[9 25] included paediatric patients, and the number of children in each were low. Therefore, it is not possible to identify factors influencing the administration of TXA to children in this review. There was also variability between the protocols and resources available to prehospital clinicians as well as settings in the included studies.

Seven of the papers reported on patients from combat settings.[25 26 29 31 33–35] Although much has been learnt about trauma care from military medicine, these studies are less likely to be generalisable to civilian trauma, and included disproportionately high numbers of young men and penetrating trauma. Despite this, the influencing factors identified were in line with those found in the civilian studies. However, in the military studies protocol and injury type seemed to play a more prominent role. Generalisability of the review was aided by the separation of civilian and military studies. Of the seven combat-specific studies, six were based on the experiences of the same prehospital service and included considerable author, data and date range overlap. A similar issue was

found for the two studies from the Netherlands.[18 19] This raises the issue of potential pseudo-replication within the review. However, strengths of this review include the extensive searches, the use of PRISMA methods and tools and the input of a multi-disciplinary study team.

Protocols played an understandably important role in TXA administration.[25] However, criteria for administration and protocol compliance varied. In general protocol compliance was poor, suggesting that alternative factors play a part in clinical decision-making. This variability in protocol compliance results in both under administration and inappropriate administration of TXA.[9 24 30 35–37] Goodwin et al found that confusion and restrictions relating to guidelines for TXA administration were barriers to administration.[22] Lipsky et al reported that 30% of TXA administrations were for patients with no clear indication.[34] They found that in some of these cases altered levels of consciousness from blunt head injury appeared to be mistaken for haemodynamic instability. McQueen and Wyse reported that 48% of TXA administrations did not fulfil the inclusion criteria in their protocol,[23] which they attributed to the influence of clinician judgement. However, a more subjective guideline relying on a clinician's ability to identify patients likely to require significant blood transfusion resulted in only 4.2% of those given TXA receiving ≥10 units of blood within the following 24 hours.[29] This suggests that clinical judgement may not be the most effective approach to guiding TXA administration, given that it can result in both low administration rates and high levels of inappropriate use.

It therefore seems preferable to focus on more objective protocols combined with improved compliance. Nadler et al found that only 22% of patients indicated for TXA in a military setting received it.[35] In a civilian setting Wong et al found that 35% of eligible patients had received prehospital TXA and reported that this was higher than other rates reported in Canada.[24] They suggested that factors such as poor intravenous accessibility, time constraints or short transport times might limit willingness to administer TXA. Nadler et al suggested that significant over and under treatment of TXA may indicate insufficient professionalism or insufficient knowledge of the practice guidelines.[35] Other papers suggested ways to improve compliance including raising awareness, improved training and the use of a decision-making tool to aid in identifying a patient's bleeding risk.[22 34] However, the inclusion of training on TXA administration is not always mandated, even within combat medical training, and compliance with training can be low.[29]

The most frequent factors mentioned in protocols were signs of clinical shock, penetrating injury and administration of other treatments. It is not surprising that patients who receive prehospital TXA tend to have more overt haemodynamic instability.[9] This may be a product of prehospital protocols that tend to be weighted towards signs of clinical shock as a key criterion for TXA administration.

The lack of obvious benefits at the time of administration was suggested as a relevant factor in the study by Goodwin et al.[22] Interestingly, Lipsky et al included the fact that survival benefit is only seen after 24–48 hours in their protocol training materials.[34] They suggest informing clinicians that they should not expect improved field haemorrhage control following TXA. It is possible that making this clear in training materials or protocols may help to address one potential barrier to TXA administration. Another factor that may contribute to low administration rates is the need for slow intravenous infusion of TXA. Clinicians in the Fisher et al study were required to administer TXA using a 10 min infusion.[29] The authors suggest this may not be compatible with a tactical situation and propose that oral or intramuscular TXA may help to increase administration rates. The requirement for slow intravenous administration was also found to be a barrier in the study by Goodwin et al.[22] Participants noted that the preparation of the drug (drawn up from two vials for an adult dose) could be a barrier in a time pressured situation.

Age was also found to influence TXA administration with younger patients generally more likely to receive TXA,[18 19 24 29 32] although the difference in age was often of questionable clinical significance. However, the evidence around a sex difference in administration was very clear.[27] Female patients are less likely to be treated with TXA in all risk categories despite TXA reducing trauma deaths to a similar extent in men and women. The disparity was greatest in older women and women at lower risk.

Following the 2019 CRASH3 study and subsequent subanalyses,[3] TXA has been indicated in the UK for adult patients with isolated traumatic head injury (GCS score of 12 or less) since July 2020.[38] However, all of the studies in this review used data predominantly predating the reporting of CRASH3. The Nutbeam et al data did briefly overlap the CRASH3 publication, but it did not specifically report on head injuries.[27] The two van Wessem papers also briefly overlapped but they specifically excluded patients with isolated head injuries.[18 19] Of the three papers looking at trauma patients with severe brain injury all three showed patient age and the severity of injury to be influencing factors, with more severely injured patients more likely to receive TXA.[17–19] However, the influence of age on this patient group was mixed and requires further study. It seems there is a lack of evidence around the factors influencing TXA administration in patients with severe head injuries and in the case of isolated head injury this may be due to the relatively recent changes to protocols around TXA administration for these patients.

Other important barriers to administration identified by Goodwin et al included a lack of knowledge and experience with TXA, a lack of resources and difficulty in identifying patients at risk of bleeding.[22] These findings may help to explain the low protocol compliance identified in this review, although none of the studies examined causal factors in detail. Injury severity played a role

in administration, and although patients who are more severely injured are more likely to receive TXA,[17 24 29 39] the measure used to assess injury severity in the included studies was often the ISS, which can only be calculated accurately following hospital assessment. Therefore, it is unclear what measure of severity prehospital clinicians are using to determine the need for TXA administration, and this may introduce additional variability.

A new scoring system to identify trauma patients at risk of bleeding has recently been published. This showed that the proportion of patients receiving prehospital TXA could be increased through the use of the Bleeding Audit Triage Trauma (BATT) score.[8] The BATT score is calculated using a weighted system including age, systolic blood pressure, GCS, respiratory rate, heart rate and whether the injury was penetrating or involved high-velocity trauma. This adds weight to the idea that clinical observations are commonly used, and potentially effective, in guiding more accurate prehospital TXA administration. A challenge that emerges is how to help prehospital clinicians identify and treat not just patients at the highest risk of bleeding following trauma, but also those at lower risk who may nevertheless benefit from TXA administration, given that the drug can be safely administered to a wide spectrum of patients with traumatic bleeding, and should not be restricted to the most severely injured.[40] The newly validated prehospital BATT score shows promise in helping to identify these lower-risk patients.[8] Treating patients with a BATT score of just 2 or more (out of 27) would result in 26 fewer deaths per 10 000 trauma patients.

## CONCLUSIONS

This review highlights a lack of high-quality research addressing the factors that influence prehospital TXA administration, particularly in children or patients with isolated head injuries. Common factors identified in this review suggest a host of system and individual-level factors that may be important in determining whether TXA is administered to trauma patients in the prehospital setting. These include: knowledge and skills; consequences and social influences; injury type (including severity, injury site and mechanism of injury); protocols; resources; priorities; patient age; patient sex. Despite the well-established benefits of TXA in trauma, and the central role of time to administration,[4 5] prehospital administration rates remain low.[9 29] In order to address this evidence-practice gap it is essential that the factors influencing prehospital administration are better understood.

### Author affiliations
[1]College of Health, Science and Society, University of the West of England, Bristol, UK
[2]Research, Audit and Quality Improvement Department, South Western Ambulance Service NHS Foundation Trust, Exeter, UK
[3]Exeter HS&DR Evidence Synthesis Centre, University of Exeter Medical School, Exeter, UK
[4]Department of Health and Social Sciences, University of the West of England, Bristol, UK
[5]Research Design Service, University Hospitals Bristol NHS Foundation Trust, Bristol, UK
[6]NIHR CLAHRC South West Peninsula, University of Exeter, Exeter, UK

**Acknowledgements** We would like to thank Professor Ian Roberts (London School of Hygiene & Tropical Medicine), Professor Tim Coats (University of Leicester) and Dr Francois-Xavier Ageron (University of Lausanne) for their assistance during this project.

**Contributors** HN acts as guarantor and participated in review conception, design and coordination, searches, paper selection, quality assessment, data extraction, data synthesis and drafted the manuscript. NS participated in searches, paper selection, quality assessment and data extraction. KK participated in conception, design, paper selection, quality assessment and data extraction. SV participated in conception, design and coordination and paper selection. JRB, MR, AB, LG and HT participated in conception and design. JTC participated in design and data synthesis. SB participated in design and undertook database searches. All authors were responsible for critical revision of the manuscript for publication and approved the final version to be published.

**Funding** This review was supported by Research Capability Funding from the South Western Ambulance Service NHS Foundation Trust and the National Institute for Health Research Applied Research Collaboration South West Peninsula (CE-18-1497/HAS-NAM-18-026). The views expressed in this publication are those of the authors and not necessarily those of the National Institute for Health Research of the Department of Health and Social Care.

**Competing interests** None declared.

**Patient and public involvement** Patients and/or the public were not involved in the design, or conduct, or reporting, or dissemination plans of this research.

**Patient consent for publication** Not applicable.

**Ethics approval** Not applicable.

**Provenance and peer review** Not commissioned; externally peer reviewed.

**Data availability statement** Data are available upon reasonable request. The datasets generated and analysed during this study are not publicly available due to participant confidentiality, but are available from the corresponding author on reasonable request.

**ORCID iDs**
Helen Nicholson http://orcid.org/0000-0002-8964-0607
Laura Goodwin http://orcid.org/0000-0002-9118-4620
Maria Robinson http://orcid.org/0000-0002-6978-8575
Jo Thompson Coon http://orcid.org/0000-0002-5161-0234
Sarah Voss http://orcid.org/0000-0001-5044-5145

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
