## [Reviewer comments · BMJ Open]

ARTICLE DETAILS

TITLE (PROVISIONAL)	Factors that influence the administration of tranexamic acid (TXA) to trauma patients in pre-hospital settings: A systematic review.
AUTHORS	Nicholson, Helen; Scotney, Natalie; Briscoe, Simon; Kirby, Kim; Bedson, Adam; Goodwin, Laura; Robinson, Maria; Taylor, Hazel; Thompson Coon, Jo; Voss, Sarah; Bengner, Jonathan

VERSION 1 – REVIEW

REVIEWER	Cornelius, Angela John Peter Smith Hospital
REVIEW RETURNED	08-Mar-2023

GENERAL COMMENTS	Great attempt to elucidate the factors limiting txa use. Have you looked at doing your own study to get more specific data to the UK?
---

REVIEWER	Chen, Li Sichuan University West China Second University Hospital
REVIEW RETURNED	03-Apr-2023

GENERAL COMMENTS	This study focused on the influencing factors of the prehospital use of tranexamic acid, systematically reviewed literature in AMED, CENTRAL and other databases from 2010 to 2020, and finally identified common factors as follows: knowledge and skills; Consequences and social impact; Injury type (including severity, injury site and injury mechanism); Treatment plan; Resources; Priorities; Patient age; Gender of patient. This paper also separately reported the use of TXA in combat environment, avoiding the use differences in different environments and different requirements, which has a certain scientific nature. The influencing factors determined by the results are consistent with those found in civil studies. This manuscript has detailed discussion and analysis on the specific influence trend of each factor, providing a breakthrough point for the subsequent promotion of the use of TXA, which is conducive to improving the use of this drug according to the actual clinical situation based on the corresponding influencing factors. This paper has good clinical significance and is worthy of publication.
---

REVIEWER	Vanderboll, Kathryn University of Michigan, Taubman Health Sciences Library
REVIEW RETURNED	07-Apr-2023

GENERAL COMMENTS	Please note: This review is limited to comments about systematic review methodology, search strategy quality, and reporting standards. I am a librarian trained in conducting evidence synthesis projects, not a content expert in the topic of this review. I very much appreciate the well executed and extensive search conducted for this review. You list the search as one of your manuscript's strengths, and I agree. I liked that you reported all your search strategies (not just the strategy from one database), and that you clarified the platform that you used to search each database and
---

	the exact dates that searching was done. Due to this careful reporting, I was able to reproduce your database searches with no trouble. I liked that you had a reason for the year limit applied, instead of - as is all too common - simply setting an arbitrary 5- or 10-year limit. I also liked that you published a protocol, and discussed deviations from the protocol. I do have questions about your PRISMA flowchart: Firstly, where did you get 34 items for the "records identified from registers" section? I'm guessing that registers is referring to ClinicalTrials.gov, but you reported finding $27 + 30 + 33 = 90$ results in your earlier search documentation about ClinicalTrials.gov. Did you not apply a date limit during your ClinicalTrials.gov searching, and were those earlier two searches (27 and 30) already represented in the 33 of the third search? If so, where did the extra 1 come from (you reported 34, not 33)? Secondly, how did you get 1781 duplicates removed before screening? In your previous search documentation, you report 605 (original search) + 189 (first search update) + 473 (second search update) + 222 (second search update deduplicated against first) = 1489 duplicates. Thirdly, in the "identification of studies via other methods" section, you list out 65 from citation searching and 29 from OpenGrey. Why were only 4 sought for retrieval? Were there some duplicates compared to what was found in the database searches? Did you have another round of title/abstract screening that's unreported in this flowchart? Furthermore, I have a quick clarification point related to the flowchart. You currently have 20 "studies included in review," and 0 "reports of included studies." My understanding of PRISMA is that "study" refers to the actual study performed, and the "report" refers to the publication that presents results of that study. A single study may produce many "reports" during and after it is conducted, with the same or overlapping datasets. So these two options in this box of the flowchart allow you to account for multiple articles published about the same study. Your manuscript could benefit from one more quick grammar review, specifically for punctuation and spacing (especially before or after citations). All in all, I applaud the high quality of searching and reporting done for this review!
--	--

VERSION 1 – AUTHOR RESPONSE

Reviewer Reports:

Reviewer: 1

Dr. Angela Cornelius, John Peter Smith Hospital

Comments to the Author:

Great attempt to elucidate the factors limiting txa use. Have you looked at doing your own study to get more specific data to the UK?

- Thank you for your positive feedback. Our team undertook the Goodwin study which looked at UK paramedic factors, however, we may look to do expand on this work in the future.

Reviewer: 2

Dr. Li Chen, Sichuan University West China Second University Hospital

Comments to the Author:

This study focused on the influencing factors of the prehospital use of tranexamic acid, systematically reviewed literature in AMED, CENTRAL and other databases from 2010 to 2020, and finally identified common factors as follows: knowledge and skills; Consequences and social impact; Injury type (including severity, injury site and injury mechanism); Treatment plan; Resources; Priorities; Patient age; Gender of patient. This paper also separately reported the use of TXA in combat environment, avoiding the use differences in different environments and different requirements, which has a certain scientific nature. The influencing factors determined by the results are consistent with those found in civil studies. This manuscript has detailed discussion and analysis on the specific influence trend of each factor, providing a breakthrough point for the subsequent promotion of the use of TXA, which is conducive to improving the use of this drug according to the actual clinical situation based on the corresponding influencing factors. This paper has good clinical significance and is worthy of publication.

- Thank you for your very positive feedback.

Reviewer: 3

Ms. Kathryn Vanderboll, University of Michigan

Comments to the Author:

Please note: This review is limited to comments about systematic review methodology, search strategy quality, and reporting standards. I am a librarian trained in conducting evidence synthesis projects, not a content expert in the topic of this review.

I very much appreciate the well-executed and extensive search conducted for this review. You list the search as one of your manuscript's strengths, and I agree. I liked that you reported all your search strategies (not just the strategy from one database), and that you clarified the platform that you used to search each database and the exact dates that searching was done. Due to this careful reporting, I was able to reproduce your database searches with no trouble.

I liked that you had a reason for the year limit applied, instead of - as is all too common - simply setting an arbitrary 5- or 10-year limit.

I also liked that you published a protocol, and discussed deviations from the protocol.

- Thank you for your positive feedback on the methodology, it is much appreciated. I

do have questions about your PRISMA flowchart:

Firstly, where did you get 34 items for the "records identified from registers" section? I'm guessing that registers is referring to ClinicalTrials.gov, but you reported finding $27 + 30 + 33 = 90$ results in your earlier search documentation about ClinicalTrials.gov. Did you not apply a date limit during your Clinical Trials.gov searching, and were those earlier two searches (27 and 30) already represented in the 33 of the third search? If so, where did the extra 1 come from (you reported 34, not 33)?

- These 34 records were from the ClinicalTrials.gov register. We did not apply a date limit and so the total number (90) included duplicates. However, for the second and third search, we found that there were both new records as well as records from previous searches that were now missing or had since been removed from the registry. Therefore, the number reported

in the PRISMA diagram (34) was the total number of unique records found over the three searches. A note of explanation has been added to the literature search report and the wording in the PRISMA diagram has been changed to read "Clinical Trials Registers" to clarify where the 34 records came from.

Secondly, how did you get 1781 duplicates removed before screening? In your previous search documentation, you report 605 (original search) + 189 (first search update) + 473 (second search update) + 222 (second search update deduplicated against first) = 1489 duplicates.

- The additional duplicates were removed by Covidence when the searches were uploaded for screening.

Thirdly, in the "identification of studies via other methods" section, you list out 65 from citation searching and 29 from OpenGrey. Why were only 4 sought for retrieval? Were there some duplicates compared to what was found in the database searches? Did you have another round of title/abstract screening that's unreported in this flowchart?

- A round of title/abstract screening was used for the studies identified via other methods. For clarity, another box for records screened and one for records excluded has been added to the "identification of studies via other methods" section of the PRISMA diagram.

Furthermore, I have a quick clarification point related to the flowchart. You currently have 20 "studies included in review," and 0 "reports of included studies." My understanding of PRISMA is that "study" refers to the actual study performed, and the "report" refers to the publication that presents results of that study. A single study may produce many "reports" during and after it is conducted, with the same or overlapping datasets. So these two options in this box of the flowchart allow you to account for multiple articles published about the same study.

- Thank you for clarifying. The PRISMA diagram has been amended to show the number of studies and reports.

Your manuscript could benefit from one more quick grammar review, specifically for punctuation and spacing (especially before or after citations).

- The manuscript has been reviewed with particular care to standardise spaces before or after citations.

All in all, I applaud the high quality of searching and reporting done for this review!

- Thank you for your positive feedback.

VERSION 2 – REVIEW

REVIEWER	Vanderboll, Kathryn University of Michigan, Taubman Health Sciences Library
REVIEW RETURNED	16-May-2023

GENERAL COMMENTS	Please note: This review is limited to comments about systematic review methodology, search strategy quality, and reporting standards. I am a librarian trained in conducting evidence synthesis projects, not a content expert in the topic of this review. I appreciate the thoroughness with which the authors addressed my comments in the previous round of review. I have no further suggestions for improvement.
---